# Characterizing the landscape of gene expression variance in humans

Scott Wolf [1], Diogo Melo [1,2]*, Kristina M. Garske [1], Luisa F. Pallares [1,2,3], Amanda J. Lea [4,5], Julien F. Ayroles [1,2]*

**1** Lewis-Sigler Institute for Integrative Genomics, Princeton University, Princeton, New Jersey, United States of America, **2** Department of Ecology and Evolutionary Biology, Princeton University, Princeton, New Jersey, United States of America, **3** Friedrich Miescher Laboratory of the Max Planck Society, Tübingen, Germany, **4** Department of Biological Sciences, Vanderbilt University, Nashville, Tennessee, United States of America, **5** Child and Brain Development, Canadian Institute for Advanced Research, Toronto, Canada

addolcet These authors contributed equally to this work.
* damelo@princeton.edu (DM); jayroles@princeton.edu (JFA)

**Data Availability Statement:** Code for reproducing all analyses and figures, along with a walk-through, is available at github.com/ayroles-lab/expressionVariance-code. All relevant data are within the paper and its Supporting Information

## Abstract

Gene expression variance has been linked to organismal function and fitness but remains a commonly neglected aspect of molecular research. As a result, we lack a comprehensive understanding of the patterns of transcriptional variance across genes, and how this variance is linked to context-specific gene regulation and gene function. Here, we use 57 large publicly available RNA-seq data sets to investigate the landscape of gene expression variance. These studies cover a wide range of tissues and allowed us to assess if there are consistently more or less variable genes across tissues and data sets and what mechanisms drive these patterns. We show that gene expression variance is broadly similar across tissues and studies, indicating that the pattern of transcriptional variance is consistent. We use this similarity to create both global and within-tissue rankings of variation, which we use to show that function, sequence variation, and gene regulatory signatures contribute to gene expression variance. Low-variance genes are associated with fundamental cell processes and have lower levels of genetic polymorphisms, have higher gene-gene connectivity, and tend to be associated with chromatin states associated with transcription. In contrast, high-variance genes are enriched for genes involved in immune response, environmentally responsive genes, immediate early genes, and are associated with higher levels of polymorphisms. These results show that the pattern of transcriptional variance is not noise. Instead, it is a consistent gene trait that seems to be functionally constrained in human populations. Furthermore, this commonly neglected aspect of molecular phenotypic variation harbors important information to understand complex traits and disease.

## Author summary

Gene expression variance, or the variation in the level of gene expression within a population, can have significant impacts on physiology, disease, and evolutionary adaptations. While the average level of gene expression is typically the focus of research, the variation

files, and at github.com/diogro/
expressionVariance-manuscript (archived at doi:
10.5281/zenodo.8028690).

**Funding:** S.W. is supported by the National
Science Foundation Graduate Research Fellowship
Program (DGE-2039656). D.M. is funded by a
fellowship from the Princeton Presidential
Postdoctoral Research Fellows Program. K.M.G. is
funded by National Institutes of Health (NIH) grant
F32ES034668. L.F.P. was funded by a Long-Term
Postdoctoral Fellowship from the Human Frontiers
Science Program and is funded by the Max Planck
Society. A.J.L. is supported by the Canadian
Institute for Advanced Research Global Scholars
Program, the Searle Scholars Program, and
through the NIH/NIGMS (R35GM147267). J.F.A. is
funded by grants from the NIH: National Institute of
Environmental Health Sciences (R01-ES029929)
and National Institute of General Medical Sciences
(NIGMS) (R35GM124881). The funders had no
role in study design, data collection and analysis,
decision to publish, or preparation of the
manuscript.

**Competing interests:** The authors have declared
that no competing interests exist.

around this average level (i.e., gene expression variance) can also be important for understanding complex traits and disease. Here, we investigate the landscape of transcriptional variance across tissues, populations, and studies. Using large publicly available RNA-seq data sets, we were able to identify the general properties associated with high- and low-variance genes, as well as factors driving variation in variance across genes. Specifically, we uncovered gene expression variance was significantly associated with gene length, nucleotide diversity, the degree of connectivity and the presence of non-coding RNA. Our results suggest that the mechanisms responsible for maintaining optimal levels of variation in high- versus low-variance differ, and that this variability is the result of different patterns of selection.

# Introduction

Molecular phenotypes such as gene expression are powerful tools for understanding physiology, disease, and evolutionary adaptations. In this context, average trait values are usually the focus of investigation, while variation around the average is often considered a nuisance and treated as noise [1]. However, gene expression variance can be directly involved in determining fitness [2, 3], can drive phenotypic variation [4], and the genetic architecture of variance itself can evolve [5]. This suggests that studying gene expression variance as a bona fide trait, its genetic architecture, and the evolutionary mechanisms shaping and maintaining gene-specific patterns of variance has the potential to further our understanding of complex traits and disease [6–9].

Variability is ubiquitous in nature and is, alongside its counterpart, robustness, a fundamental feature of most complex systems. But, at the same time, the degree of variability seems to differ between genes [1] suggesting that it might be associated with biological function and therefore be shaped by selection. From a mechanistic perspective, several competing forces act to shape transcriptional variance [5, 10], and the outcome of the interaction between these processes is still poorly understood [11]. For example, we expect the influx of new mutations to increase the variance, while the selective removal of these polymorphisms, via purifying selection or selective sweeps, to decrease it [12, 13]. From a quantitative trait perspective, stabilizing selection should decrease variation around an optimal value, and directional selection can lead to a transient increase in variance while selected alleles sweep to fixation, followed by a reduction in variance as these alleles become fixed. Pleiotropic effects are also important, as they allow selection on one trait to influence the variance of other traits [14, 15]. Both indirect effects of directional selection on variance open the possibility that the main driver of gene expression variance is not direct selection on variance but indirect effects due to selection on trait means [11]. How the interaction of these processes shape gene expression variance is an open question. However, some general predictions can be made. If a homogeneous pattern of stabilizing selection is the main driver of gene expression variance, we would expect transcriptional variance to be consistent regardless of the population, tissue, or environmental context. If idiosyncratic selection or environmental patterns are more important, we could observe large differences in gene expression variance across studies.

A key difficulty in addressing these questions is that the constraints on gene expression variance might also be dependent on the gene tissue specificity. Mean expression is known to differ across tissues [16], however, to what extent differential expression (i.e., differences in mean expression level) translate into differences in expression variance is not clear. Higher mean expression could lead to higher variance, but other processes can also affect transcriptional

variance. For example, if a gene is expressed in more than one tissue and variance regulation is independent across tissues, stabilizing selection on gene expression could be more intense depending on the role of that gene in a particular tissue, causing a local reduction in variation that leads to differences in variance across tissues (Fig 1A). These across-tissue differences would not necessarily follow mean expression. Alternatively, expression variation across tissues could be tightly coupled and, in this example, selection in one tissue would lead to a reduction in variance across tissues, resulting in a consistent pattern of variation (Fig 1B). While we lack a clear picture of how tissue-specific gene expression variation is regulated, Alemu et al. [17] used microarray data from several human tissues to show that epigenetic markers were linked to gene expression variation and that these markers were variable across tissues and between high- and low-variance genes.

To explore the landscape of gene expression variance and the association between transcriptional variance and biological function, we use 57 publicly available human gene expression data sets spanning a wide range of experimental contexts and tissues. By comparing the gene expression variance measured across such heterogeneous data sets, we show that the degree of expression variance is indeed consistent across studies and tissues. We use the observed similarities to create an across-study gene expression variance ranking, which orders genes from least variable to most variable. We then integrate various genomic-level functional annotations as well as sequence variation to probe the drivers of this variance ranking. Finally, we explore the link between gene expression variance and biological function by leveraging gene ontology and other gene annotations.

## Results

### Data sets

We use 57 publicly available human gene expression RNA-seq data sets which were derived from the publications listed in Table 1, and a complete metadata table for each study is available in S1 Dataset. We only use data sets that fulfilled the following conditions: samples came from bulk RNA-seq (and no single cell approaches), data sets were associated with a publication, sample-level metadata was available, and the post-filtering sample size was greater than 10 (note that we did not include data from non-baseline/exposure/stimulated datasets). Our focus is on bulk RNA-seq data sets because we are interested in studying population level variation. In particular, we are interested in variation due to segregating genetic variants, which is expected to contribute to the evolution of gene expression, and studies using bulk RNA tend to have many more genetically variable biological replicates. These data sets span 13 different tissue types and the post-filtering mean sample size we used for each data set was 390, with a median of 251, and ranged from 12 to 2931 samples. Several data sets were derived from two large consortia: GTEx [16] and TCGA [18], and we note the origin of the data sets in the figures where appropriate. We refer to data sets and studies interchangeably, and so each tissue in GTEx is referred to as a different study. The final list of genes used from each study can be found in S2 Dataset.

### What level of variation are we exploring?

Variance, variation, or variability? There is no consensus in the literature as to when to refer to these different levels of variation and under what scenario. The type of variation that we refer to here as *gene expression variance* or *transcriptional variance* has also been called gene expression variability or even gene expression noise. Following [19], we suggest that variation and variability should be distinguished, and variation should be used to refer to the realized differences across some population at any organization scale. For example, a gene will have high

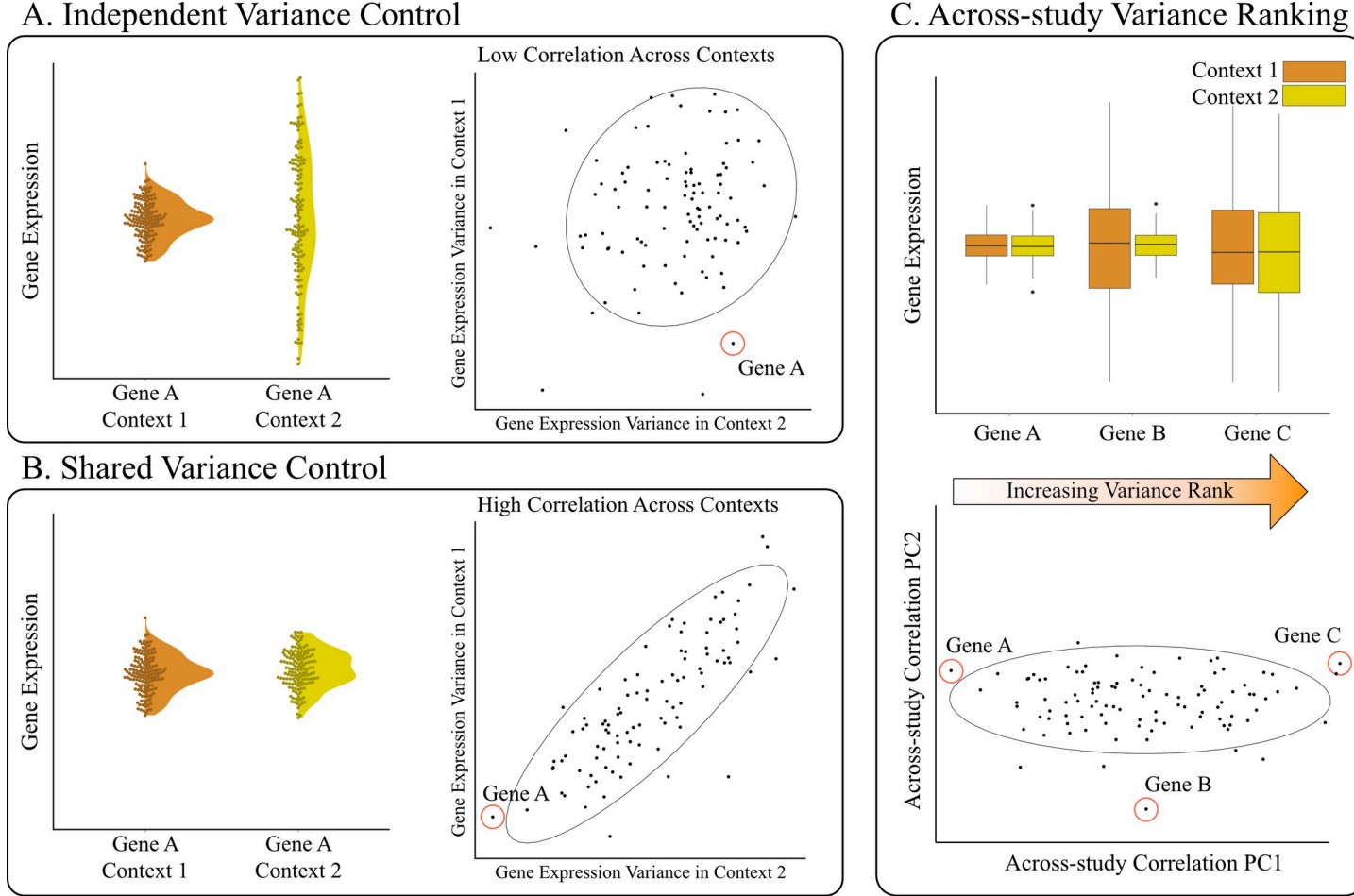

**Fig 1. Example of how differences in the regulation of transcriptional variance can drive changes in the correlations between gene expression variance measures.** In (A), independent regulation causes the reduction in variation to be restricted to context 1 (context here can refer to different tissues, environments, populations, studies, etc.). On the right side of panel A, independent regulation results in low correlation across contexts. In (B), a shared regulatory architecture maintains consistent variance across both conditions, leading to high similarity in transcriptional variance across contexts. In (C), we see how the similarity seen in panel B can be leveraged to create an across-context rank of gene expression variance. When transcriptional variance ranks are highly correlated, the rank of the projection onto the first principal component (PC1) allows us to summarize the across-context pattern of transcriptional variance. Note that genes can have intermediate rank because they have intermediate expression variance, or because their gene expression variance is not consistent across conditions. The higher the variance explained by PC1, the more we expect intermediate genes to be consistent.

variation if its gene expression differs greatly across individuals (or cells, or conditions, etc.), leading to a high transcriptional variance (the statistical measure of variation). We would then specify at which level of variation we are working with: across individuals, across cells, within genotypes, and so on. In contrast, variability should refer to the potential to generate variation. For example, a given gene might have low realized variation because all individuals in a population have a similar genotype and environment, which leads to similar expression levels across individuals. At the same time, this same gene can also have a large mutational target for its expression level, leading to many potential mutations that would cause its expression level to change, and therefore we would consider this gene to have high variability and low variation. It should be clear that in this manuscript we are interested in population level, across individual, realized variation in gene expression, measured as gene expression variance. Since we are using phenotypic variation, the variance we measure is a combination of the contribution of

**Table 1. Data set source references.** Columns show the study ID, with the corresponding tissue in parentheses, and the source publication.

| Study ID | Citation |
| --- | --- |
| ADIPOSE_TISSUE (Fat), ADRENAL_GLAND (Adrenal), BLOOD (Blood), BLOOD_VESSEL (Blood_vessel), BONE_MARROW (Marrow), BRAIN (Neuron), HEART (Heart), BREAST (Breast), SALIVARY_GLAND (Salivary), COLON (Colon), LIVER (Liver), NERVE (Neuron), LUNG (Lung), PANCREAS (Pancreas), MUSCLE (Muscle), THYROID (Thyroid), OVARY (Ovary), STOMACH (Stomach), ESOPHAGUS (Esophagus), SPLEEN (Spleen), PROSTATE (Prostate), SKIN (Skin), PITUITARY (Pituitary), TESTIS (Testis) | The GTEx Consortium, 2020 –[40] |
| LUSC (Lung), STAD (Stomach), COAD (Colon), LUAD (Lung), BRCA (Breast), KIRC (Kidney), KIRP (Kidney), LIHC (Liver), THCA (Thyroid), PRAD (Prostate), UCEC (Uterus) | The Cancer Genome Atlas Research Network et al., 2013 –[18] |
| SRP150552 (Blood) | Altman et al., 2019 –[41] |
| SRP101294 (Fat) | Armenise et al., 2017 –[42] |
| SRP057500 (Platelets) | Best et al., 2015 –[43] |
| SRP051848 (Immune) | Breen et al., 2015 –[44] |
| SRP187978 (Liver) | Çalışkan et al., 2019 –[45] |
| E-ENAD-34 (Immune) | Chen et al., 2016 –[46] |
| SRP059039 (Blood) | DeBerg et al., 2018 –[47] |
| SRP174638 (Immune) | Dufort et al., 2019 –[48] |
| E-GEOD-57945 (Colon) | Haberman et al., 2014 –[49] |
| SRP162654 (Blood) | Harrison et al., 2019 –[50] |
| SRP095272 (Blood) | Jadhav et al., 2019 –[51] |
| SRP102999 (Blood) | Kuan et al., 2017 –[52] |
| SRP145493 (Immune) | Kuan et al., 2019 –[53] |
| E-GEUV-1 (Immune) | Lappalainen et al., 2013 –[54] |
| SRP035988 (Skin) | Li et al., 2014 –[55] |
| SRP192714 (Blood) | Michlmayr et al., 2020 –[56] |
| ERP115010 (Blood) | Roe et al., 2020 –[57] |
| E-ENAD-33 (Neuron) | Schwartzentruber et al., 2018 –[58] |
| SRP181886 (Neuron) | Srinivasan et al., 2020 –[59] |
| SRP098758 (Blood) | Suliman et al., 2018 –[60] |
| SRP032775 (Blood) | Tran et al., 2016 –[61] |
| SRP069212 (Liver) | Yang et al., 2017 –[62] |

several sources, spanning genetic (e.g., due to segregating expression quantitative trait loci (eQTL) or variance QTL (vQTL)) and environmental differences across individuals.

## Gene expression variance

For each study, transcriptional variance per gene was measured as the standard deviation (SD) of the distribution of gene expression values for all individuals in a particular study. Mean and variance are known to be correlated in RNA-seq data, both due to the nature of count data and the expectation that more highly expressed genes should have more variation. As our focus here is on variance, we control for both of these expected drivers of transcriptional variation. To achieve this, SD was calculated using a unified pipeline that normalized the mean-variance relation in read-count data, controlled for batch effects, and removed outliers (see Methods for details, and the calculated values for means and standard deviations are available in S3 Dataset). The observed range of gene expression SDs across genes is variable but can be normalized so that the distributions are comparable (Fig 2D). This comparison reveals differences in the

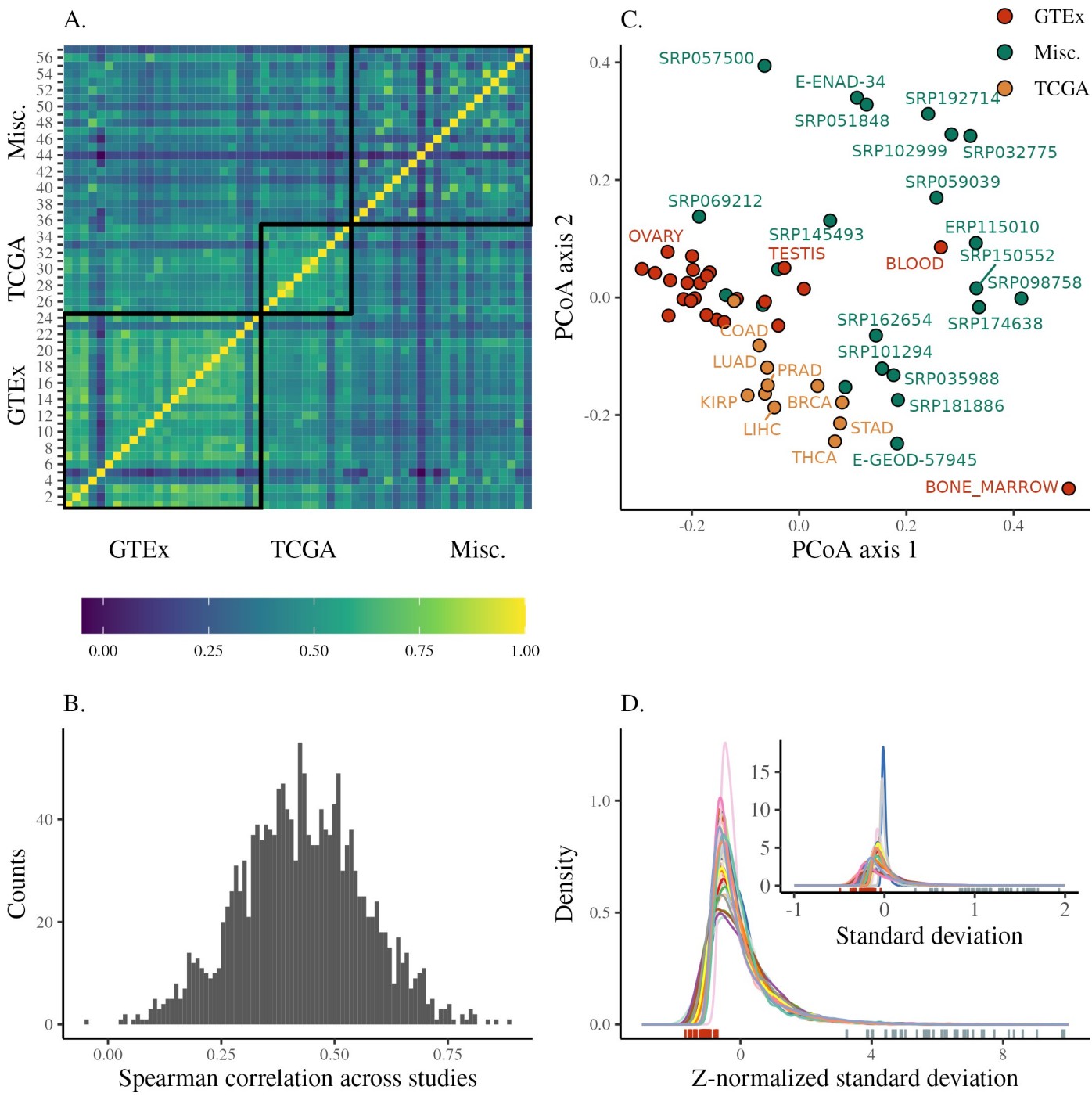

**Fig 2. Overview of the distribution of transcriptional variance across studies.** (A) Heatmap showing the correlation in transcriptional variance across studies (as the Spearman correlation of gene expression standard deviations). Pairs of studies with more similar patterns of gene expression variance have higher correlations. Studies are shown in the same order as in S1A Fig. (B) Distribution of the pairwise Spearman correlations between studies shown in the previous panel. (C) PCoA using the distance between studies derived from the pairwise correlations. (D) Density plot of standard deviations after z-normalization. Each distribution is colored by study. The inset plot shows the distribution of mean-centered standard deviations grouped and colored by study, without normalization. The corresponding rug plots show the location of the highest-ranking gene in standard deviation rank (HBB) (right, blue) and lowest (WDR33) (left, red).

range of gene expression SDs that can be due to any number of methodological or biological differences between the data sets. We avoid having to deal with these global differences in the range of variation by using only the ranking of the genes according to their gene expression SD in each study. Therefore, patterns of transcriptional variance were compared across studies using Spearman correlations ($\rho_s$) between gene expression SDs. This comparison reveals a broadly similar rank of gene expression variance as the correlations across studies are mostly positive and high (75% of correlations are between 0.45 and 0.9, Fig 2A and 2B), indicating that genes that are most variable in one study tend to be most variable in all studies. A principal coordinate analysis [20] using $|1 - \rho_s|$ as a between-study distance measure does not show clearly delineated groups, but GTEx and TCGA studies are clustered among themselves and close together (Fig 2C). This clustering indicates some effect of study source on the similarity between gene expression SD across studies, which we explore in detail below.

To characterize what factors may explain differences in across-study similarity, we directly modeled the across-study correlations using a mixed-effect linear model designed to account for the non-independence in pairwise correlation data [21, 22]. In this model (see Methods), we use a random effect for individual study ID, a fixed effect for pairwise tissue congruence (whether a comparison is within the same tissue or different tissue), and a fixed effect for pairwise study source (which pair of sources among GTEx, TCGA, and miscellaneous is involved in a comparison) as predictors of the correlations (see Methods). This model (S1 Fig) shows that comparisons of studies within GTEx and TCGA have on average higher values of $\rho_s$, but also that comparing studies across GTEx and TCGA also shows a mild increase in the average correlation (S1C Fig). Correlations that do not involve studies from TCGA and GTEx (marked as "Misc.") are on average lower (S1C Fig). While we do not have a clear explanation for this pattern, since TCGA and GTEx are independent, this mild effect on the similarities could be due to the level of standardization of the data coming from these two large consortia. Tissue type also affects the degree of similarity in transcriptional variance, with studies using the same tissue being, on average, more similar (S1B Fig). However, all these pairwise effects are mild, and the largest effects on the correlations are those associated with individual studies, in particular some specific tissues, i.e., comparisons involving BONE MARROW (from GTEx) and study SRP057500 (which used platelets) are on average lower (S1A Fig). The only negative correlation we observe is between these two studies, which also appear further away in the PCoA plot in Fig 2C.

## Transcriptional variance rank

The strong correlations between transcriptional variance across studies suggest that variance rank is indeed a property of genes that can be robustly estimated. To estimate this gene-level rank, we devised an across-study approach that allowed us to rank individual genes according to their degree of transcriptional variance by averaging the ordering across all studies. We do this by calculating the score of each gene on the first principal component of the across-study Spearman correlation matrix shown in Fig 2A. This procedure is illustrated in Fig 1C. Ordering genes using these scores generates a ranked list of genes, with the most variable genes having the highest rank. The position in the SD distributions shown in Fig 2D of the most and least variable genes in this rank illustrates how the extremes of the rank are indeed some of the least and most variable genes across all studies. In addition, to be able to account for any residual effect of mean expression on the variance, we created a similar across-study rank for mean expression. To explore tissue-specific divers or transcriptional variation, we created a set of tissue-specific SD ranks. To that end, we used the same procedure outlined above but using only

studies that were performed on the same tissue. Both tissue-specific and across-study ranks are available in S4 Dataset.

## Biological function explains gene-level transcriptional variance

As a first step toward explaining the factors that drive variation in variability between transcripts, we focused on the top 5% most variable and the bottom 5% least variable genes in the across-study ranking (560 genes in each group). A Gene Ontology (GO) enrichment analysis shows 59 enriched terms in the low-variance genes, and 738 enriched terms in the high-variance genes (using a hypergeometric test and a conservative Benjamini-Hochberg (BH) adjusted p-value threshold of $10^{-3}$; see S5 Dataset for a complete listing).

Among the most variable genes, we observe enrichment for biological processes such as immune function, response to stimulus, maintenance of homeostasis, and tissue morphogenesis (S2A Fig). Furthermore, we see a 7.7-fold enrichment for genes that encode secreted proteins in the most variable genes, relative to all other genes (hypergeometric test, $p < 10^{-3}$).

Among the least variable genes, we see enrichment for housekeeping functions such as mRNA processing, cell cycle regulation, methylation, histone modification, translation, transcription, and DNA repair (S2B Fig); accordingly, we also find a 2.0-fold enrichment in previously characterized human housekeeping genes [23] (hypergeometric test, $p < 10^{-3}$). The genes exhibiting the lowest variance are also enriched for genes that have been previously shown to have a high probability of being loss-of-function intolerant (pLI) [24] (1.2-fold enrichment, hypergeometric test, $p < 10^{-3}$). Genes with a high pLI have been shown to be important in housekeeping functions and have higher mean expression values across a broad set of tissues and cell types [24]. The observation that genes with low variance are enriched for both housekeeping genes and genes with high pLI is consistent with this previous report; and we further see that the mean expression of genes positively correlates with pLI (partial Spearman correlation $\rho_s = 0.32$, $p < 10^{-3}$), showing the opposite relationship between variance and mean expression when considering pLI.

In the previous analysis, we explored the relationship between transcriptional variance and function by starting from the extremes of the variance distribution and searching for GO enrichment among these high- and low-variance genes. We also approach the problem from the opposite direction, starting from the genes associated with each GO term and searching for enrichment for high- or low-variance genes among them, as this allows to probe the link between variation and function across the full variance rank. To this end, we gathered all biological process GO terms in level 3 (i.e., terms that are at a distance of 3 from the top of the GO hierarchy). Using level-3 terms gives us a good balance between number of terms and genes per term. We separated the genes associated with at least one of these level-3 terms into expression variance deciles, with the first decile having the lowest variance. We then counted how many genes in each decile have been associated with each term. If variance rank is not associated with the GO annotations, terms should have an equal proportion of genes in each decile. We measured how far from this uniform allocation each term is by measuring the Shannon entropy of the proportion of genes in each decile. Higher entropy is associated with a more uniform distribution of genes across deciles. GO terms with low entropy indicate some deciles are over-represented in the genes associated with that term. We also measured skewness for each term, which should be zero if no decile is over-represented, negative if high-variance terms are over-represented, and positive if low-variance deciles are over-represented. The relation between skewness and entropy for each GO term can be seen in Fig 3 and in S6 Dataset. Positive-skew low-entropy terms, those enriched with low-variance genes, are associated with housekeeping functions, like RNA localization, translation initiation, methylation, and chromosome segregation (Fig 4A).

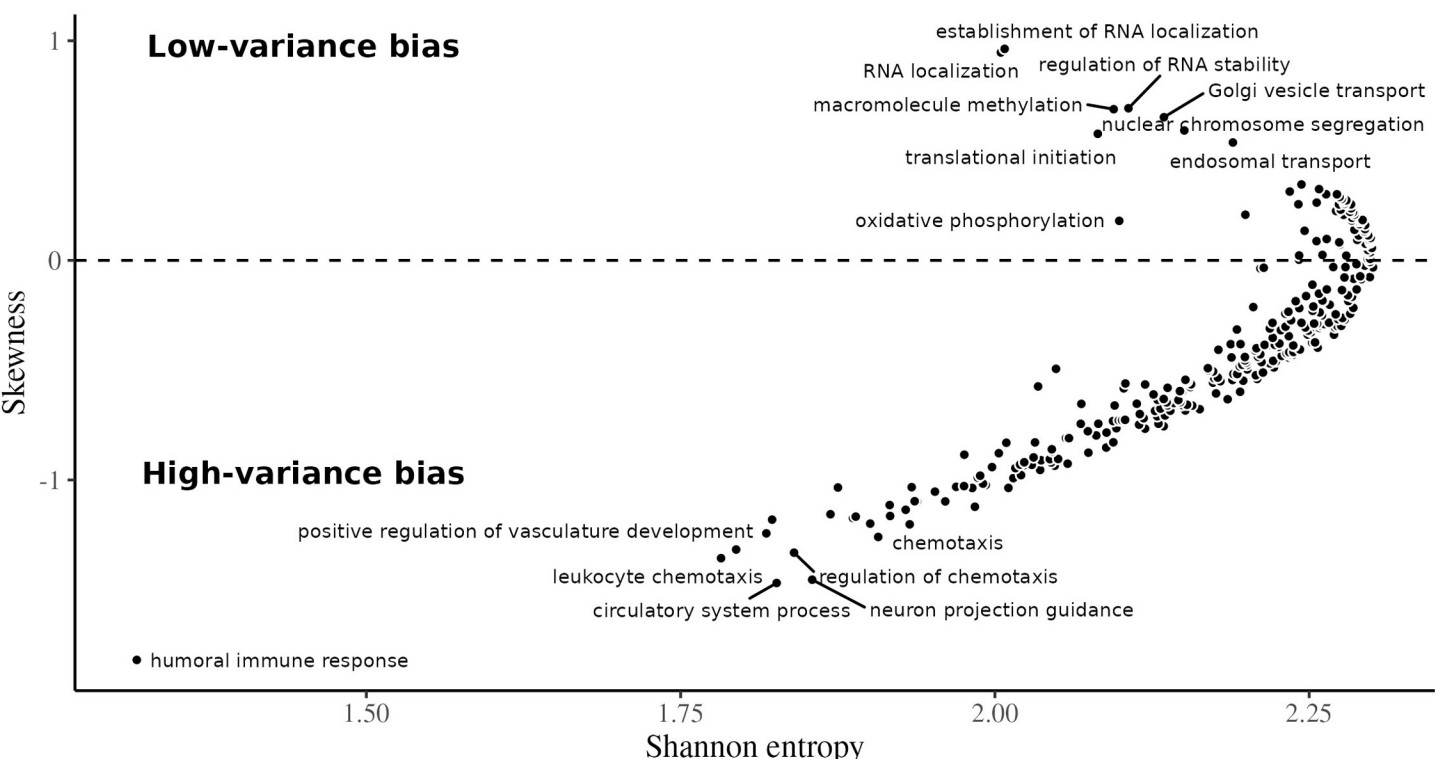

**Fig 3. Relationship between skew and entropy of rank decile distributions for each GO term.** High entropy terms, to the right of the plot, are associated with a more egalitarian proportion of genes in each of the SD rank deciles. The terms on the left of the plot are associated with more genes in some particular decile. The skewness in the y-axis measures if the high- or low-variance deciles are more represented for a particular term. Terms on the positive side of the y-axis are associated with low-variance genes, and terms on the negative side of the y-axis are associated with high-variance genes. The GO terms are filtered for gene counts greater than 100, as in Fig 4. Some of the top high- and low-skewness terms are labeled for illustration.

Likewise, terms with negative skew and low entropy, enriched for high-variance genes, are related to immune response, tissue morphogenesis, chemotaxis—all dynamic biological functions related to interacting with the environment (Fig 4B).

Both GO analyses suggest a strong association between biological function and the degree of transcriptional variance. Genes associated with baseline fundamental functions, expected to be under strong stabilizing selection, are also low-variance; high-variance genes are associated with responding to external stimuli (i.e., tissue reorganization and immune response).

## Environmental sensitivity predicts transcriptional variance

As suggested by our GO enrichment analyses, one mechanism that may generate consistent variability in gene expression is the response to environmental inputs. In other words, high-variance genes may be those that are environmentally sensitive, while low-variance genes may be robust to environmental stimuli or perturbations (or alternatively, responsive to all stimuli, such that they are always highly expressed across individuals). To understand the relationship between environmental sensitivity and variance, we drew on gene expression data from a recently generated catalog of environmentally responsive genes in lymphoblastoid cell lines (LCLs). This catalog was generated by exposing 544 LCLs derived from individuals included in the 1000 Genomes Project to each of 11 in vitro exposures (including immune signaling molecules, hormones, and man-made chemicals), as well as a control; these manipulations of the cellular environment were followed by mRNA-seq and differential expression analyses

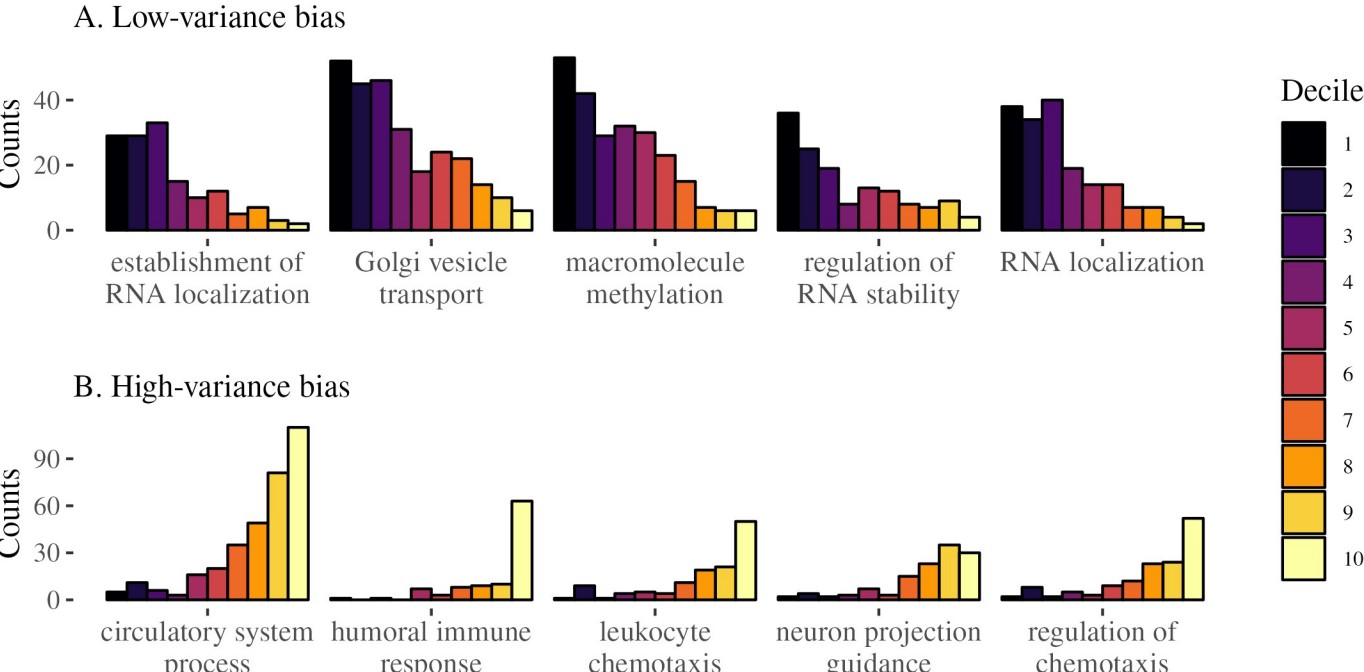

**Fig 4.** Distributions of decile ranks of level-3 GO terms. Each plot shows the count of genes in each decile of the rank. Only GO terms that are associated with at least 100 genes are used. We sort these terms by the skewness of the distribution. The top panel (A) shows the 5 most positively skewed terms, and the bottom panel (B) shows the 5 most negatively skewed terms.

comparing each treatment to its control [25]. Using lists of environmentally responsive genes derived from this study, we found that high-variance genes were more likely to respond to at least one in vitro exposure, relative to genes not classified as high- or low-variance (Fisher's exact test: $p < 0.05$, odds = 1.524); as predicted, the same is not true for low-variance genes (Fisher's exact test: $p = 0.993$, odds = 0.797). When we analyzed each exposure separately, we found that high-variance genes were more likely to be responsive to 4 out of the 10 environments we explored (1 environment was dropped due to a lack of differentially expressed genes in the original experiment; Fisher's exact test, FDR < 10%; S2 Table). These exposures included key immune stimuli and hormones such as interferon gamma and dexamethasone (a synthetic glucocorticoid). In contrast, we found that low-variance genes showed the opposite pattern: they are significantly underrepresented among environmentally responsive genes across 4/10 environments (Fisher's exact test, FDR < 10%; S2 Table). Though not all of our environment-specific tests reached statistical significance, it is also worth noting that almost all 10 environments showed concordance in effect size direction (i.e., high-variance genes tended to be overrepresented among environmentally responsive genes and low-variance genes tended to be underrepresented). While the above analyses show that high-variance genes tend to overlap with genes induced by a given exposure, we hypothesized that genes that are similarly induced by many different exposures may, in fact, exhibit moderate or low variance. In other words, genes induced by many stimuli may always be highly expressed across individuals, and thus low variance, while genes induced by select stimuli may only be upregulated in a subset of the population, and thus exhibit high variance. In support of this idea, we found that, among genes that responded to at least one environment in the LCL experiment, high-variance genes responded to a median of only one environment, while both low-variance genes and the background set responded to a median of 4/10 environments (generalized linear model

comparing high-variance to background and low-variance, $p < 10^{-7}$ and $p < 10^{-11}$, respectively). Thus, high-variance genes are indeed more likely to be environmentally sensitive, but in a highly select and stimulus-specific manner, which we hypothesize drives their between-individual heterogeneity. We note that all analyses presented in this section focused on the composite set of high- and low-variance genes defined across tissues, but we obtain similar results when focusing on blood, the tissue in our dataset most similar to LCLs (S2 Table).

## Evolutionary forces at play in shaping transcriptional variance

We use three gene-level summary statistics, nucleotide diversity ($\pi$), gene expression connectivity, and the rate of adaptive substitutions ($\alpha$), as a proxy to assess whether selection might be involved in shaping gene expression variance. For all the correlations in this section, we use partial Spearman correlations that include the mean gene expression rank as a covariate, which accounts for any residual mean-variance correlation (Spearman correlation between mean and SD ranks, $\rho_s$ = -0.07, $p < 10^{-18}$). Nucleotide diversity in the gene region is used as a proxy for the impact of cis-regulatory genetic variation on transcriptional variance. As expected, low-variance genes tend to have lower levels of polymorphisms (partial Spearman correlation, $\rho_s$ = 0.184, $p < 10^{-87}$). Gene-gene connectivity, a proxy for gene regulatory interactions and selective constraints [26], is, in turn, negatively correlated with the expression variance (partial Spearman correlation, $\rho_s$ = -0.08, $p < 10^{-21}$), supporting the expectation that highly connected genes are more constrained in their variation. Finally, we also find that low-variance genes tend to have fewer substitutions by comparing the across-study rank with $\alpha$ (partial Spearman correlation, $\rho_s$ = -0.044, $p < 10^{-2}$), in line with the expectation that genes under stronger selection should be less variable. Despite all associations being significant and in the expected direction, their effect sizes are very small, suggesting a weak link between these broad measures and transcriptional variance.

## Specific gene regulatory signatures are associated with transcriptional variance

To assess how local epigenetic features relate to gene expression variance we calculate the proportion of the gene ($\pm 10$ kb) that corresponds to epigenetic signatures of gene regulation defined through ChromHMM [27] chromatin states. Chromatin states associated with distal (i.e., non-promoter) gene regulation are positively correlated with the across-study variance rank, regardless of whether the regulatory effect on gene expression is positive or negative (Fig 5; see across-study correlations in S3A Fig). For example, both the proportion of gene regions made up of enhancers and repressed genomic states are positively correlated with gene expression variance (BH adjusted Spearman correlation, $p < 0.05$). In contrast, histone modifications associated with active promoters, as well as transcribed states, are inversely correlated with gene expression variance (S3A Fig), whereas they are positively correlated with the mean rank (S3B Fig). Taken together, these results are compatible with gene expression variance being regulated through distal (i.e., non-promoter) gene regulatory mechanisms, rather than the overall active transcriptional state of a gene region, as is the case with mean gene expression.

Given that ChromHMM chromatin states are available for specific tissues, we asked whether the regulatory signatures associated with the across-study variance rank are recapitulated at the tissue level. Many of the across-study correlations are recapitulated at the tissue-specific level (with two exceptions noted below), including a strong and highly consistent positive correlation between the proportion of gene regions made up of enhancer states and that gene's expression variance, and an inverse relationship between gene expression variance and histone marks associated with gene transcription (S3A Fig). Two blood associations stand out

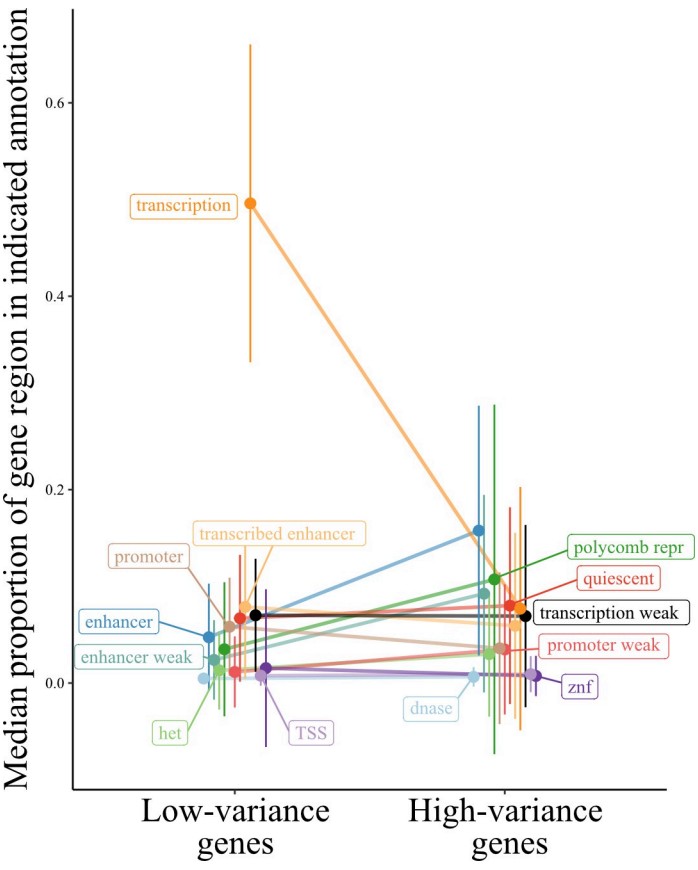

**Fig 5. Proportion of gene regions made up of ChromHMM chromatin states for low- and high-variance genes.**
The line plot contrasts the proportion of gene regions made up of the indicated chromatin states for genes in the top and bottom 5% of the across-study variance rank metric. Ends denote the median proportion of gene regions made up of the chromatin state, and error bars represent the standard error of the mean. States colored black are not significant, all others exhibit significant differences between low- and high-variance genes (BH adjusted Wilcoxon signed-rank test, p < 0.05). Het indicates heterochromatin; TSS, transcription start sites; znf, zinc finger genes. The mean rank version of this analysis is shown in S4 Fig.

as being different from the consistent effects across the other tissue-level and across-study associations. First, the weak (i.e., histone marks associated with both activating and repressive functions) promoter state is positively correlated with transcriptional variance in all comparisons except blood. Second, the consistent inverse correlation of gene expression variance with weak transcription is reversed in blood, such that there is a positive correlation between histone marks associated with weak transcription and blood gene expression variance (S3A Fig). Taken together, these results suggest that, rather than genes with a bivalent promoter state (i.e., poised genes) exhibiting more expression variance, blood high-variance genes are more likely already expressed at basal levels (i.e., weakly transcribed), as discussed previously [28].

Immediate early genes (IEGs) respond quickly to external signals without requiring *de novo* protein synthesis, and a bivalent state has been reported to be associated with IEG promoters [reviewed in 29]. Given our results that genes with high expression variance are enriched for cellular signaling and response mechanisms (S2A Fig), and bivalent promoter states are correlated with the gene expression variance rank (S3A Fig), we hypothesized that IEGs would be enriched within genes in the top expression variance ranks. This was the case for all tissue-level gene expression variance ranks (enrichment ratios range from 3.3–8.8, Bonferroni-

adjusted hypergeometric test, p < 0.05), except for blood (enrichment ratio = 1.2, hypergeometric test, p = 0.3). Thus, once again, blood stands out when attempting to understand genomic regulatory drivers of expression variance. In all, while high-variance genes are generally shared across tissues and enriched for immune and environmental signaling pathways, it seems that the gene regulatory mechanisms governing their expression are distinct between immune cell types and other tissues studied here.

### Linking expression variance and disease

To explore the link between transcription variance and genes known to be associated with human diseases, we used a data set designed to provide causal relationships between gene expressions and complex traits/diseases (based on a probabilistic transcriptome-wide association study (PTWAS) [30]). Using the list of significant gene-disease pairs at 5% FDR provided by Zhang et al. [30], we performed a hypergeometric enrichment test for the top 5% high- and low-variance genes in our across-study rank and in all tissue-specific gene variance ranks. We use both across-study and tissue-specific ranks because some genes only appear in the tissue-specific rank due to their limited tissue-specific gene expression. In the high-variance group, we find no enrichment in the across-study rank, but we do find enrichment of genes annotated for allergy, immune disease, and endocrine system disease among the high-variance genes in several tissue-specific variance ranks. For example, among high-variance genes in the colon, we see enrichment for endocrine system disease (1.77-fold, hypergeometric test, $p < 10^{-4}$). Among high-variance genes in the immune cells, we see enrichment for endocrine system disease (1.67-fold, hypergeometric test, $p < 10^{-3}$), allergy (1.7-fold, hypergeometric test, $p < 10^{-3}$), and immune disease (1.32-fold, hypergeometric test, $p < 10^{-2}$). Among high-variance genes in the thyroid, we see enrichment for endocrine system disease (1.9-fold, hypergeometric test, $p < 10^{-5}$), allergy (1.85-fold, hypergeometric test, $p < 10^{-4}$), and immune disease (1.45-fold, hypergeometric test, $p < 10^{-4}$). These are all rather similar and suggest a stable pattern of high-variance gene expression across these tissues, with enrichment for these three classes of diseases. The link with immune diseases is expected given the high enrichment for immune-related genes in the high-variance group [8]. As for the low-variance group, we found strong enrichment for genes associated with psychiatric and neurological disorders in the across-study rank and in some tissue-specific ranks (breast, liver, and stomach; ∼ 1.2-fold enrichment, hypergeometric test, p < 0.05, for all cases). The psychiatric disease link is consistent with previous work [7] and is discussed below; however, the enrichment among the low-variance genes is weaker.

### Discussion

Using large publicly available data sets allowed us to probe the landscape of transcriptional variance in humans. We find a broadly similar pattern of transcriptional variance, evidenced by the high correlations between gene expression variance across most studies, consistent with measurements of expression variance for various tissues [6, 17]. Leveraging this similarity between gene expression variance across tissues and contexts, we developed a multivariate strategy to create a single rank of expression variance, which allowed us to order almost 13k genes (∼65% of the genes expressed in humans) according to their transcriptional variance. Using this rank, we were able to study the general properties associated with high- and low-variance genes as well as factors driving variation in variance across genes.

Some differences in gene expression variance were driven by technical aspects of gene expression measurement (with data derived from large consortia showing more similar patterns of variance across genes), and by tissue (with studies using the same tissues also showing

higher similarities). This suggests that careful consideration of sample sizes and experimental design are fundamental to the study of gene expression variance, and the usual small samples of RNA-seq studies might be underpowered for the study of this particular aspect of gene expression. In particular, comparing studies within GTEx was associated with higher similarities in gene expression variance rank. This higher similarity could be due to some individuals being included in more than one GTEx study, and therefore the same eQTLs could be driving similarities in gene expression variance in these samples. However, both the effects of study origin and tissue were small, and the largest drivers of differences across studies were idiosyncratic differences related to single data sets, with tissues known to have divergent gene expression patterns (i.e., bone marrow, blood, testis, and platelets) also showing the largest differences in gene expression variance. Understanding the consequences of these differences in variance for specific tissues is still an open field. Also, because we are working with bulk samples, a part of these differences could be due to cell-type heterogeneity across samples [31]. Furthermore, variation in single cell expression has also been linked to cell-type function [9]. In summary, it is clear, that differences in variance are informative beyond the differences in mean expression. Even after we account for differences in mean expression, differences in gene expression variance carry information about tissue origin and function.

Functional analyses using GO enrichment indicated a clear link between function and gene expression variance. On the one hand, genes with high transcriptional variance were enriched for biological functions related to response to environmental stimuli, such as immune function and tissue reconstruction. On the other hand, low-variance genes were enriched for basic cell functions, (e.g., RNA processing, translation, DNA methylation, and cell duplication). These results are consistent with previous analyses of gene expression variance on a tissue-by-tissue basis [17]. This pattern of enrichment is also observed when we look at enrichment for high- or low-variance genes within the genes associated with each term in the GO hierarchy. Basic cell function terms are enriched for low-variance genes, and terms involved in response to external stimulus are enriched for high-variance genes.

While indirect, all these patterns point to a selective structuring of gene expression variance. Stabilizing and purifying selection are consistent: genes expected to be under strong stabilizing selection, those linked with fundamental baseline biological processes, are indeed overrepresented in the least variable genes. These same genes are also expected to be under strong purifying selection and to show low levels of polymorphisms, which we observe. Likewise, genes whose function is constrained by myriad interactions with several other genes, those with high connectivity, are less variable. Furthermore, genes involved with direct interaction with the environment, which must change their pattern of expression depending on external conditions, are expected to be more variable, and again we see a strong enrichment of environmentally responsive genes among the most variable. Given this strong link between function and variance, it is not surprising that the gene variance ranking is similar across data sets.

One interesting aspect of the GO term analysis shown in Figs 3 and 4 is that there is no GO biological process term associated with enrichment for intermediate variance genes: the low-entropy terms have either positive or negative skew, never zero skew. In other words, there is no annotated biological process for which the associated genes are kept at some intermediary level of variation. For the GO terms we used, either there is no relation between the transcriptional variance and the biological process, or there is a strong bias toward high or low-variance genes. This suggests that selective shaping of gene expression has two modes, corresponding with (1) biological processes under strong stabilizing selection (i.e., variance-reducing selection) or (2) biological processes under disruptive selection (i.e., variance-increasing selection). The only outlier in the skewness-by-entropy relation is the "oxidative phosphorylation" term, which displays relatively low entropy and low skewness, and indeed we see that there is some

enrichment for intermediate variance rank in the genes associated with this term (S6 Fig). In short, we find strong support for the idea that there are genes with consistently more (or less) variable expression levels, and that these differences in variance are the result of different patterns of selection.

Following Alemu et al. [17], we observe that epigenetic signatures of gene regulation, such as enhancer histone marks, make up a higher proportion of the surrounding genomic regions of genes that exhibit higher variance in expression. In contrast, an accumulation of strong promoter elements and overall transcriptional activation is associated with genes with lower expression variance. These results suggest the presence of distinct modes of regulation for genes with high vs. low variance. Combined, the differences in the types of genomic regulatory features surrounding the high- and low-variance genes and their distinct functional annotations suggest different mechanisms of regulation of their gene expression variance [17]. This heterogeneity could lead to detectable differences in selection signatures between distal regulatory elements and promoters, depending on the transcriptional variance. This heterogeneity in regulation for high and low-variance genes suggests that important biological information has been overlooked given the focus that the field has placed on understanding gene expression robustness, in the sense of reducing variation [32–35]. For example, Siegal and Leu [32] provide several examples of known regulatory mechanisms for reducing gene expression variance, but no examples for the maintenance of high gene expression variance. We posit that it should be possible to go beyond the usual characterization of mechanisms of gene expression robustness, in the sense of reducing variation, and to explore mechanisms for the *robustness of plasticity*, that is, the maintenance of high levels of gene expression variation given environmental cues.

Given the broad consistency of gene expression variance in healthy tissues, a natural question is how do these well-regulated levels of variation behave in disease conditions. We find some suggestive links between tissue-specific variance ranks and disease, but these links need to be further explored using more specific methods. Comparing two HapMap populations, Li et al. [6] showed that gene expression variance was similar in both populations and that high-variance genes were enriched for genes related to HIV susceptibility, consistent with our observation of enrichment for immune-related genes among those with more variable expression. In a case-control experiment, Mar et al. [7] showed that expression variance was related to disease status in Schizophrenia and Parkinson's disease patients, with altered genes being non-randomly distributed across signaling networks. These authors also find a link between gene network connectivity and expression variance, in agreement with the effect we find using the gene expression variance rank. The pattern of variance alteration differed across diseases, with Parkinson's patients showing increased expression variance, and Schizophrenia patients showing more constrained patterns of expression. The authors hypothesize that the reduced variance in Schizophrenia patients reduces the robustness of their gene expression networks, what we refer to as a loss of plasticity. This suggests that several types of shifts in gene expression variation are possible, each with different outcomes. We highlight three distinct possibilities: First, low-variance genes, under strong stabilizing selection, could become more variable under stress, indicating a reduced capacity for maintaining homeostasis. Second, high-variance genes, expected to be reactive to changes in the environment, could become less variable, indicating a reduced capacity to respond to external stimuli. Third, the covariance between different genes could be altered, leading to decoherence between interdependent genes [36]. Any one of these changes in expression variance patterns could have physiological consequences, and exploring these differences should be a major part of linking gene expression to cell phenotypes and function (see Hagai et al. [8] for example). Genes are also expected to differ in their capacity to maintain an optimal level of gene expression variance [34]. Variation in robustness is linked to gene regulatory networks and epigenetic gene expression regulation

[33, 37] and, therefore, should differ across high- and low-variance genes. Our results suggest that the mechanisms responsible for maintaining optimal levels of variation in high- and low-variance could differ and that this variability is the result of different patterns of selection.

## Methods

### Data sources

We selected 57 human RNA-seq data sets from the public gene expression repositories recount3 [38] and Expression Atlas [39]. We only used data sets with an associated publication, for which raw read count and sample-level metadata were available. Because we are interested in individual-level variation of gene expression, we exclude single-cell studies. Metadata and details on the included data sets can be found in the S1 Dataset. We use the word "studies" to refer to independent data sets, which could have been generated by the same consortium. For example, the GTEx data are separated by tissue, and we refer to each tissue as a separate study. We divide our data sets into three categories depending on their origin: GTEx, TCGA, and Miscellaneous.

### Processing pipeline

We use a standardized pipeline to measure gene expression variance while removing extraneous sources of variation. Because we are interested in variation under non-perturbed conditions, data from case-control studies were filtered to keep only control samples. Technical replicates were summed. For each study, we filtered genes that did not achieve a minimum of 1 count per million (cpm) reads in all samples and a mean of 5 cpm reads across samples. To account for library size and the mean-variance relation in RNA-seq count data, we applied a variance stabilizing transformation implemented in the function vst from the DESeq2 R package [63] to the genes passing the read-count filters. This mean-variance correction was verified by plotting mean-variance relations before and after correction, and these plots can be seen in S1 Appendix. Some studies still show higher variances for genes with lower expression even after the vst correction, like BLOOD, COLON, and STAD, but given that these do not show a large difference in the variance rank when compared to the other studies, we do not expect this residual mean effect to be relevant. Various technical covariates (like experimental batch, sex, etc.) were manually curated from the metadata associated with each study and accounted for using an independent linear fixed-effects model for each study. A list of covariates used for each study is available in S1 Dataset. Outlier individuals in the residual distribution were removed using a robust Principal Component Analysis (PCA) approach of automatic outlier detection described in [64]. This procedure first estimates robust Principal Components for each study and then measures the Mahalanobis distance between each sample and the robust mean. Samples that are above the 0.99 percentile in Mahalanobis distance to the mean are marked as outliers and removed. We verify that the batch effect correction and outlier removal are reasonable by using PCA scatter plots after each step of the pipeline to check the result for residual problems like groupings or other artifacts. These PCA plots before and after batch correction and outlier removal are also included in S1 Appendix. After all sample filtering, the mean sample size we used for each data set was 390, with a median of 251, and ranged from 12 to 2931 samples. Gene expression standard deviations (SDs) are measured as the residual standard deviations after fixed effect correction and outlier removal. We choose standard deviation as a measure of variation to have a statistic on a linear scale, and we do not use the coefficient of variation because we have already corrected for mean differences and for the mean-variance relation inherent to RNA-seq count data [1].

## Correlations in transcriptional variance

We assessed the similarity in gene expression variance across studies by using an across-study Spearman correlation matrix of the measured SDs. Only genes present in all studies were used to calculate the Spearman correlation matrix, $\sim 4200$ genes in total. To do this, the SDs for this common subset of genes are organized in a $N \times \rho$ matrix $D$, where $N$ is the number of genes and $\rho$ the number of studies. Then, we take the Spearman correlation between all the columns of $D$, resulting in a symmetric $\rho \times \rho$ correlation matrix $\boldsymbol{\rho}$ (shown in Fig 2A). Using Spearman correlations avoids problems related to overall scaling or coverage differences, and allows us to assess if the same genes are usually more or less variable across studies.

To investigate the factors affecting the correlations between studies, we used a Bayesian random effects model to estimate the effect of study origin and tissue on the correlations across studies. This model is designed to take the non-independent nature of a set of correlations into account when modeling the correlation between gene expression SDs. This is accomplished by adding a per-study random effect, see [22] for details. For each pair of studies $i$ and $j$, the Fisher z-transformed Spearman correlation between their SDs ($z(\boldsymbol{\rho}_{ij})$) is modeled as:

$$
\begin{aligned}
z\left(\boldsymbol{\rho}_{ij}\right) &\sim N\left(\mu_{ij}, \sigma\right), \text{ for all } i > j \\
\mu_{ij} &= \mu_0 + \alpha_i + \alpha_j + \beta_{[t_{ij}]} + \gamma_{[so_{ij}]} \\
\alpha_i &\sim N(0, \sigma_\alpha), \text{ for } i = 1 \ldots 57 \\
\gamma_k &\sim N(0, 1/4), \text{ for } k = 1 \ldots 6 \\
\beta_l &\sim N(0, 1/4), \text{ for } l = 1, 2 \\
\mu_0 &\sim N(0, 1) \\
\sigma, \sigma_\alpha &\sim Exp(1)
\end{aligned}
$$

The $\alpha_i$ terms account for the non-independence between the pairs of correlations and estimate the idiosyncratic contribution of each study to all the correlations it is involved in. We measure the effects of tissue congruence and study-origin congruence on the correlations using the fixed effect coefficients $\beta_l$ and $\gamma_k$. The $\beta_l$ coefficients measure the effect of studies $i$ and $j$ sharing the same tissue (index variable $t_{ij} = 1$) or not ($t_{ij} = 2$). The $\gamma_k$ coefficients measure the effect of comparing studies with the same or different study-origins, and the index variable $so_{ij}$ codifies all six of the unique pairwise combinations between GTEx, TCGA, and Misc (index variable $so_{ij} = 1$ implies a comparison between $i$ in GTEx and $j$ in GTEx, $so_{ij} = 2$ would be a comparison between GTEx and Misc, and so on). We also explored a version of this model that included the effect of sample size on the pairwise correlations, but sample size did not have a relevant effect and so was dropped in the final model. All fixed effect parameters ($\beta_l$ and $\gamma_k$) receive weakly informative normal priors with a standard deviation of one quarter and a mean of zero. This choice of prior implies a plausible range of -1 to 1 for each of the fixed effects coefficients, which is appropriate given that the range of $z(\boldsymbol{\rho}_{ij})$ is approximately 0 to 1.5. For the residual standard deviation ($\sigma$) and the random effects standard deviation ($\sigma_\alpha$), we use an exponential prior with a rate of one, and for the intercept ($\mu_0$), a normal prior with a mean of zero and a standard deviation of one. This model was fit in Stan [65] via the *rethinking* R package [66], using eight chains, with 4000 warm-up iterations and 2000 sampling iterations per chain. Convergence was assessed using R-hat diagnostics [67], and we observed no warnings or divergent transitions.

## Gene expression SD rank

Given that most of the variation in the Spearman correlation across studies is explained by a single principal component (PC1 accounts for 62% of the variation in the across-study Spearman correlation matrix, while PC2 accounts for only 5%; see S5 Fig), we use the ranked projections of gene expression SDs in this principal component (PC1) to create an across-study rank of gene variation. The higher the rank, the higher the expression SD of a given gene. Genes that were expressed in at least 50% of the studies were included in the rank. To project a particular gene onto the PC1 of the across-study correlation matrix, we impute missing values using a PCA-based imputation [68]. The imputation procedure has minimal effect on the ranking and imputing missing SD ranks at the beginning or at the end of the ranks produces similar results. We also create a tissue-specific variance ranking, using the same ranking procedure but joining studies done in the same tissue type. For this tissue-level ranking, we only use genes that are expressed in all studies of a given tissue, and in this case, no imputation is required. For tissues that are represented by a single study, we use the SD ranking for that study as the tissue rank.

## Gene expression mean rank

We also use the same strategy to create a mean gene expression rank, repeating the process but using mean expression instead of standard deviation. All ranks are available in S4 Dataset.

## Gene level statistics

**Genetic variation.** Genetic variation measures were obtained from the PopHuman project, which provides a comprehensive set of genomic information for human populations derived from the 1000 Genomes Project. Gene-level metrics were used when available. If only window-based metrics are available, we assembled gene-level information from 10 kb window tracks where each window that overlaps with a given gene was assigned to the gene and the mean metric value is reported. In parallel, we use the PopHumanScan data set, which expands PopHuman by compiling and annotating regions under selection. Similarly, we used gene-level information when possible, and for tracks with only window-based metrics, gene-level information was assembled from the 10 kb windows using the same assignment method described above. Nucleotide diversity ($\pi$), the average pairwise number of differences per site among the chromosomes in a population [69], provides insight into the genetic diversity within a population, in this case, the CEU population within 1000 genomes.

**Gene connectivity.** For each data set, we calculated the weighted connectivity for all genes by creating a fully connected gene-by-gene graph in which each edge is weighted by the Spearman correlation between gene expression levels across samples. We then trimmed this graph by keeping only edges for which the Spearman correlation is significant at a BH false discovery rate of 1%. In this trimmed network, we then took the sum of the Spearman correlation of all remaining edges for each gene. So, for each study, we have a measure of the total correlation of each gene with every other gene. The weighted connectivity for each gene is the average across all studies in which that gene is expressed.

**Cross-tissue vs. tissue-level chromatin states.** We use the universal [70] and tissue-specific [71] ChromHMM [27] chromatin states to compare the non-overlapping genome segmentation to cross-tissue and tissue-level gene expression variance metrics. We use the proportion of the gene regions (gene ± 10 kb) made up of each of the ChromHMM chromatin states.

**Correlations.** We use the ppcor R package v1.1 [72] to calculate the pairwise partial Spearman correlations between gene-level statistics and the gene expression variance rank while

controlling for the mean expression rank. P-values are corrected using the Benjamini-Hochberg procedure and comparisons with an adjusted $p < 0.05$ are considered significant.

## Gene function assessment

**GO term enrichment.**   All gene ontology (GO) analyses were done using the clusterProfiler R package v4.2.2 [73] and the Org.Hs.eg.db database package v3.14.0 [74]. GO and all further enrichment analyses used the hypergeometric test to assess the significance of the enrichment. Level-3 GO terms used in the skewness by entropy analysis can be found in S6 Dataset, along with the estimated skewness and entropy values.

**Environmentally responsive genes.**   We used the list of environmentally responsive genes available in the supporting information from Lea et al. [25]. When we overlapped the list of LCL-expressed genes with our list of ranked genes, we retained 9282 genes in the cross-tissue analyses presented in the main text, and 5574 genes in the blood-specific analyses presented in S1 Table. We used Fisher's exact tests to ask whether high-variance genes were more likely to be responsive to >0 environments relative to all genes not included in the low-variance category (and vice versa for low-variance genes). We also used Fisher's exact tests followed by Benjamini-Hochberg false discovery rate correction to ask whether high-variance genes were more likely to be responsive to each individual environment relative to all genes not included in the low-variance category (and vice versa for low-variance genes). Finally, we used generalized linear models with a Poisson error structure to test for an effect of gene category (high-variance, low-variance, or neither) on the number of environments that a gene responded to (ranging from 0–11).

**Housekeeping genes.**   Human housekeeping genes were identified as genes that are expressed with low variance in all 52 human cell and tissue types, assessed in over 10,000 samples [23]. We test for enrichment of housekeeping genes in the genes within the highest and lowest 5% of gene expression variance rank.

**Probability of being loss-of-function intolerant (pLI).**   Genes that are likely haploinsufficient (i.e., intolerant of heterozygous loss-of-function variants) were detected as those with fewer than expected protein-truncating variants (PTVs) in ExAC [24]. We use genes with a pLI > 0.9 to test for the enrichment of loss-of-function intolerant genes in the genes exhibiting the highest and lowest 5% gene expression variance estimates.

**Secreted genes.**   We use The Protein Atlas [75] to extract information on which proteins are secreted [76] and test for enrichment of genes with secreted products in the genes within the highest and lowest 5% of gene expression variance rank.

**Immediate early genes (IEGs).**   Human IEGs were curated from the literature in [77] as genes that respond to experimental stimulation through up-regulation within the first 60 minutes of the experiment. We use the hypergeometric test to assess the significance of the enrichment. Immediate early genes (IEGs): Human IEGs were curated from the literature in [77] as genes that respond to experimental stimulation through up-regulation within the first 60 minutes of the experiment.

**Disease annotations.**   We use the gene annotations for involvement with diseases provided by the supporting information S2 Table from Zhang et al. [30] and test for enrichment for disease annotations in the genes within the highest and lowest 5% of gene expression variance rank.

## Supporting information

**S1 Fig. Modeling the correlations between transcriptional variance across studies.**
(PDF)

**S2 Fig. GO enrichment analysis of the most and least variable genes.**
(PDF)

**S3 Fig. Across-study and tissue-specific gene expression variance and mean correlations with non-overlapping chromatin states through ChromHMM.**
(PDF)

**S4 Fig. Proportion of gene regions made up of ChromHMM chromatin states for genes in the top and bottom 5% of the across-study mean rank metric.**
(PDF)

**S5 Fig. Scree plot showing variance explained by each PC of the across-study Spearman correlation matrix of gene expression standard deviations.**
(PDF)

**S6 Fig. Distributions of decile ranks of the GO term oxidative phosphorylation.**
(PDF)

**S1 Table. Variance and mean rank metrics and the corresponding ChromHMM annotations used.**
(PDF)

**S2 Table. Enrichment analysis of environmentally responsive genes in LCLs.**
(XLSX)

**S1 Appendix. Diagnostics plots for processing pipeline.**
(PDF)

**S1 Dataset. Study metadata.** Metadata file describing the data used in the study as well as some intermediate processing information.
(CSV)

**S2 Dataset. Study gene lists.** List of genes included in each study after filtering.
(CSV)

**S3 Dataset. Gene expression means and standard deviations.** Tables with final calculated means and standard deviations.
(XLSX)

**S4 Dataset. Gene ranks.** Gene expression mean and variance ranks, across-study and tissue-specific.
(XLSX)

**S5 Dataset. GO enrichment.** Combined table describing gene ontology enrichment in the top 5% and bottom 5% of genes as ranked by variance.
(CSV)

**S6 Dataset. Level-3 GO terms.** Skewness by Shannon entropy for level-3 GO terms.
(CSV)

## Acknowledgments

We thank all members of the Ayroles lab for their support. We thank Noah Rose and Cara Weisman for their thoughtful comments. We thank Pedro Madrigal for help with the Expression Atlas interface. We also acknowledge that the work reported in this paper was

substantially performed using the Princeton Research Computing resources at Princeton University which is a consortium of groups led by the Princeton Institute for Computational Science and Engineering (PICSciE) and Office of Information Technology's Research Computing.

## Author Contributions

**Conceptualization:** Scott Wolf, Diogo Melo, Luisa F. Pallares, Julien F. Ayroles.

**Data curation:** Scott Wolf, Diogo Melo, Kristina M. Garske.

**Formal analysis:** Scott Wolf, Diogo Melo, Kristina M. Garske, Amanda J. Lea.

**Funding acquisition:** Scott Wolf, Diogo Melo, Kristina M. Garske, Luisa F. Pallares, Amanda J. Lea, Julien F. Ayroles.

**Investigation:** Scott Wolf, Diogo Melo, Kristina M. Garske, Amanda J. Lea.

**Methodology:** Scott Wolf, Diogo Melo, Kristina M. Garske, Luisa F. Pallares, Amanda J. Lea, Julien F. Ayroles.

**Project administration:** Julien F. Ayroles.

**Resources:** Julien F. Ayroles.

**Software:** Scott Wolf, Diogo Melo, Kristina M. Garske.

**Supervision:** Julien F. Ayroles.

**Validation:** Scott Wolf, Diogo Melo.

**Visualization:** Scott Wolf, Diogo Melo, Kristina M. Garske.

**Writing – original draft:** Diogo Melo.

**Writing – review & editing:** Scott Wolf, Diogo Melo, Kristina M. Garske, Luisa F. Pallares, Amanda J. Lea, Julien F. Ayroles.

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
