## [Decision Letter · Decision Letter 0]

2 Apr 2023

Dear Dr Melo,

Thank you very much for submitting your Research Article entitled 'Characterizing the landscape of gene expression variance in humans' to PLOS Genetics.

The manuscript was fully evaluated at the editorial level and by independent peer reviewers. The reviewers appreciated the attention to an important problem, but raised some substantial concerns about the current manuscript. Based on the reviews, we will not be able to accept this version of the manuscript, but we would be willing to review a much-revised version. We cannot, of course, promise publication at that time.

If you decide to revise the manuscript for further consideration at PLOS Genetics, please aim to resubmit within the next 60 days, unless it will take extra time to address the concerns of the reviewers, in which case we would appreciate an expected resubmission date by email to plosgenetics@plos.org.

We are sorry that we cannot be more positive about your manuscript at this stage. Please do not hesitate to contact us if you have any concerns or questions.

Yours sincerely,

James J Cai

Guest Editor

PLOS Genetics

Gregory Barsh

Editor-in-Chief

PLOS Genetics

Reviewer's Responses to Questions

**Comments to the Authors:**

Reviewer #1: Summary: This paper examines the landscape of variance in gene expression using various public bulk RNA-seq data. It ranks genes according to their variance and systematically examines the association between high or low variance genes and their function and epigenetic signature.

The study provides important information on gene expression variance. The study is well designed and the manuscript is well written. On the other hand, the interpretation of the results is sometimes inadequate or misleading. I recommend the publications of this manuscript after the revision.

Major comment

The relevant part of the comment in the text is written in italics.

Comment1:

What is the difference between the analysis using Skeveness and entoropy shown in Fig. 3 and the Gene Ontology Enrichment analysis? Both appear to search for gene sets associated with high or low variance gene sets. For why did you perform these two different analyses?

Comment2:

(Result) However, all these pairwise effects are mild, and the largest effects on the correlations are those associated with individual studies, in particular some specific tissues, i.e., comparisons involving BONE MARROW (from GTEx) and study SRP057500 (which used

platelets) are on average lower (SI fig. 1 A).

Recent studies have shown that cellular subset heterogeneity affects the variance of bulk gene expression (https://www.sciencedirect.com/science/article/pii/S2001037022003968). For example, when high and low gene expressing subsets are mixed in a sample, the dispersion of the cellular subset fraction between samples affects the gene expression variance.

Considering the diversity of immune cell subsets and their individual differences, It seems this result is reasonable and not surprising. How can these results be interpreted in terms of cellular subset heterogeneity?

Comment3:

> (Discussion) We find a broadly similar pattern of transcriptional variance, evidenced by the high correlations between gene expression variance acrossmost studies, consistent withmeasurements of expression variance in single cells and in populations of cells for various tissues [6,16,29].

I think this sentence is misleading. expression variance in bulk samples and "cell-to-cell variability" quantified in single cell studies are different concept (See the second paragraph in Discussion Section in (https://www.sciencedirect.com/science/article/pii/S2001037022003968). This is because in a bulk experiment, the "cell-to-cell variability" information is averaged out. It is not surprising that they share a common cause, but this difference should be clarified and discussed to prevent readers from misunderstanding.

Minor comment

Comment1: In Fig2(D), It is unclear what the many coloured lines represent; please specify in Legend.

Reviewer #2: The authors examine the variance of gene expression in 4200 expressed genes collected in 57 studies, including GTEx and TCGA. "Study" here refers to study-tissue combination, so for example GTEx contributes multiple studies (one per tissue) all deriving from the same "study origin". For each study, gene expression variance is calculated for each gene in a way that decorrelates from that gene's overall mean expression. The gene variances are ranked within study, and those ranks are used to compare gene variances across studies, so as to identify genes that are consistently high variance or low variance regardless of study. These results are then used to identify trends gene variance across tissues and in relation to gene function, and this provides the basis for discussion about evolutionary aspects of variance with regard to stabilizing selection vs dynamic biological function. In particular, lower gene variance is seen for house keeping genes whereas higher variance is seen for genes involved in responsive processes.

The methods and analyses used are appropriate, mostly clear, and in many cases innovative. The arguments made by the authors seem mostly well-supported by the data, and the manuscript is for the most part well written and persuasive.

I have some concerns about the message of the manuscript and the descriptions of the methods, and a number of minor comments.

1. The Methods section "Correlations in transcriptional variance" needs more detail and explanation. Lines 398-410 describe a Bayesian

"varying effects" model, by which the authors presumably mean a Bayesian "random effects" or "mixed" model (and if so then should amend this), to model the Fisher transform of the correlations rho_ij. The authors do not define i and j but need to. Indeed, it would benefit this section to use notation earlier, eg, define the G x C matrix D containing elements s_gc, the normalized standard deviation of gene g=1,...,G in study i=1,...,C, and so on. The model specification needs more detail and some clarifications. The variance parameter of the normal description is written as sigma; better to stick with convention and call this sigma^2. It is not stated whether you fit the model to rho_ij for {i,j: i > j} or all i and j, or what. The design matrix X measures "the effects of tissue congruence and study-origin congruence" but the predictors should be stated clearly, eg, two indicator variables, one for same tissue and the other for same study-origin, where "study origin" is also defined explicitly. The priors could also be defined more precisely: is the alpha prior centered at zero? Is a unit normal prior what most people call a standard normal N(0,1), and unit exponential the same as Exp(1)?

2. There has recently been increasing interest in genetic and non-genetic effects on phenotypic variance that are separate from effects on phenotypic means (eg, PMID:20585554, PMID:21467569, PMID: 23150607, PMID:33326753, PMID:30389794), often called phenotypic variability or variance heterogeneity, but also variance. In light of those studies, it is important to emphasize that this study is not about gene expression variability. For example, within a population with a mean-controlling eQTL at high MAF, the gene expression variance could in theory entirely be due to that eQTL. Similarly, the reason why genes have similar expression variance ranks between studies from the same study origin (eg, between different tissue studies in GTEx) could entirely be due to the fact they were collected on the same genetic population, whose gene expression variance could be determined by the same eQTL effects on the expression means. Therefore, when the authors describe gene expression variance as being separate from effects of the mean, it will be good to emphasize (more than once) that the variance being studied includes variance due to segregating eQTLs. It might even be useful to have a table describing the sources of gene expression variance and their explanation -- eQTLs, temporal fluctuations, environment effects, etc.

3. Figure C upper panel was confusing me initially. Underneath the horizontal axis it says "Increasing Variance Rank" but the variances in context 2 are not monotonically increasing. Please clarify this figure.

4. Line 108: it would be useful to report the correlation between the mean ranks and the std ranks, which is notable in that it is neglible.

5. Line 434-438. In the calculation of gene connectivity, the authors trim insignificant edges to each gene, then calculate the average correlation weight of the remaining edges. Does this mean that a gene with ten significant edges of average weight 0.4 will score as less connected than a gene with two significant edges of weight 0.5?

6. Line 148 / Figure 3: What is the GO term at around (2.05, 0.08)? This one sticks out by having both low entropy and low skewness, suggesting a U-shaped distribution of std dev deciles. It would be good to include the data points for this plot in SI data 5 (I couldn't find it).

7. Line 376-377: "[DESeq2's vst] mean-variance correction was verified by plotting mean-variance relations before and after correction [plots in SI]". The authors use DESeq2's vst() function to remove the mean-variance relationship. That function applies an estimated variance stabilizing function that is generally fine but, because of its log-based construction, can work poorly for genes with low mean read counts, as seen for blood, colon, and stad. The authors should point this out and--assuming this is the case--reassure the reader overall conclusions are not affected.

**Have all data underlying the figures and results presented in the manuscript been provided?**

Reviewer #1: Yes

Reviewer #2: Yes

PLOS authors have the option to publish the peer review history of their article (what does this mean?). If published, this will include your full peer review and any attached files.

Reviewer #1: No

Reviewer #2: No

---

## [Decision Letter · Decision Letter 1]

11 Jun 2023

Dear Dr Melo,

Thank you very much for submitting your Research Article entitled 'Characterizing the landscape of gene expression variance in humans' to PLOS Genetics.

The manuscript was fully evaluated at the editorial level and by independent peer reviewers. The reviewers appreciated the attention to an important topic but identified some concerns that we ask you address in a revised manuscript.

We therefore ask you to modify the manuscript according to the review recommendations. Your revisions should address the specific points made by each reviewer.

Yours sincerely,

James J Cai

Guest Editor

PLOS Genetics

Gregory Barsh

Editor-in-Chief

PLOS Genetics

Please address the remining comments from Reviewer 2. Otherwise, congratulations for the publication of a fine paper.

Reviewer's Responses to Questions

**Comments to the Authors:**

Reviewer #1: All my concerns have been addressed. This paper is OK.

Reviewer #2: The authors have done a thorough job addressing my previous comments.

Major comment:

The items listed in the manuscript's Supporting Information are not easily cross-referenced with the files in the corresponding github repository. I strongly suggest that, in addition to the github, the authors put the 15 SI items in some other kind of permanent storage (eg, Figshare, Dryad, etc) so readers can easily locate them.

Minor comment:

Line 446. The normal priors with variance 1/4 are described as weakly informative. Perhaps this counts as weak information, but without knowing the variance of rho_ij, 1/4 seems like it might impose a lot of shrinkage. Perhaps in parentheses, the authors could state the variance (or sd) of the rho_ij values, just to put the 1/4 (sd=1/2) in context.

**Have all data underlying the figures and results presented in the manuscript been provided?**

Reviewer #1: Yes

Reviewer #2: **No: **The items listed in the manuscript's Supporting Information are not easily cross-referenced with the files in the corresponding github repository. I strongly suggest that, in addition to the github, the authors put the 15 SI items in some other kind of permanent storage (eg, Figshare, Dryad, etc) so readers can easily locate them.

PLOS authors have the option to publish the peer review history of their article (what does this mean?). If published, this will include your full peer review and any attached files.

Reviewer #1: No

Reviewer #2: No

---

## [Editor Report · Decision Letter 2]

15 Jun 2023

Dear Dr Melo,

We are pleased to inform you that your manuscript entitled "Characterizing the landscape of gene expression variance in humans" has been editorially accepted for publication in PLOS Genetics. Congratulations!

Yours sincerely,

James J Cai

Guest Editor

PLOS Genetics

Gregory Barsh

Editor-in-Chief

PLOS Genetics

Comments from the reviewers (if applicable):

**Data Deposition**

http://datadryad.org/submit?journalID=pgenetics&manu=PGENETICS-D-23-00013R2

**Press Queries**

---

## [Editor Report · Acceptance letter]

2 Jul 2023

PGENETICS-D-23-00013R2 

Characterizing the landscape of gene expression variance in humans 

Dear Dr Melo, 

We are pleased to inform you that your manuscript entitled "Characterizing the landscape of gene expression variance in humans" has been formally accepted for publication in PLOS Genetics! Your manuscript is now with our production department and you will be notified of the publication date in due course.

With kind regards,

Zsofia Freund

PLOS Genetics

On behalf of:
